# Interacting effects of vegetation components and water level on methane dynamics in a boreal fen

Terhi Riutta[1,2,3], Aino Korrensalo[4], Anna M. Laine[4], Jukka Laine[1] and Eeva-Stiina Tuittila[1,4]

[1] University of Helsinki, Department of Forest Ecology, Helsinki, Finland
[2] Current address: University of Oxford, School of Geography and the Environment, Oxford, UK
[3] Imperial College London, Department of Life Sciences, Ascot, UK
[4] School of Forest Sciences, University of Eastern Finland, Finland

*Correspondence to:* Eeva-Stiina Tuittila (eeva-stiina.tuittila@uef.fi)

**Abstract.** Vegetation and hydrology are important controlling factors in peatland methane dynamics. This study aimed at investigating the role of vegetation components, sedges, dwarf-shrubs, and *Sphagnum* mosses, in methane fluxes of a boreal fen under natural and experimental water level drawdown conditions. We measured the fluxes during growing seasons 2001-2004 using static chamber technique in a field experiment where the role of the ecosystem components was assessed via plant removal treatments. The first year was a calibration year after which the water level drawdown and vegetation removal treatments were applied. Under natural water level conditions, plant-mediated fluxes comprised 68-78% of the mean growing season flux ($1.73 \pm 0.17$ g $CH_4$ m$^{-2}$ month$^{-1}$ from June to September), of which *Sphagnum* mosses and sedges accounted for 1/4 and 3/4, respectively. The presence of dwarf shrubs, on the other hand, had a slightly attenuating effect on the fluxes. In water level drawdown conditions, the mean flux was close to zero ($0.03 \pm 0.03$ g $CH_4$ m$^{-2}$ month$^{-1}$) and the presence / absence of the plant groups had a negligible effect. In conclusion, water level acted as a switch; only in natural water level conditions vegetation regulated the net fluxes. The results are relevant for assessing the response of fen peatland fluxes to changing climatic conditions, as water level drawdown and the consequent vegetation succession are the major projected impacts of climate change on northern peatlands.

**Keywords:** climate change, dwarf shrubs, methane, peatland, sedges, *Sphagnum*

## 1. Introduction

Approximately one-third of all terrestrial carbon is stored in boreal and subarctic peatlands (e.g. Yu, 2012) that generally act as $CO_2$ sinks in current climatic conditions. However, pristine wetlands, including peatlands, marshes and floodplains, are also the largest natural source of methane ($CH_4$) into the atmosphere (Ciais et al., 2014; Kirschke et al., 2013; Saunois et al., 2016). The carbon sink function of peatlands is mostly due to the slow decomposition rate resulting from waterlogged, anaerobic conditions sustained by a high water level, which simultaneously favour $CH_4$ production. $CH_4$ is the end product of anaerobic decomposition by strictly anaerobic methanogenic archaea. It is released from the

peat into the atmosphere via diffusion through the peat column, ebullition or plant-mediated transport (Lai, 2009). A considerable part, from 20 to up to 90% (Le Mer and Roger, 2001; Pearce and Clymo, 2001; Whalen, 2005) of the $CH_4$ diffusing through the upper, aerobic part of the peat layer is oxidized to $CO_2$ by methanotrophic bacteria (MOB) before reaching atmosphere.

Vegetation is a major factor controlling peatland $CH_4$ fluxes (Koelbener et al., 2010; Ström et al., 2005, 2012). Fresh root litter and exudates are important substrates for the methanogenic microbes, and a significant proportion of the $CH_4$ is formed from this easily available organic matter instead of from old, recalcitrant peat (Koelbener et al., 2010; Ström et al., 2012). Therefore, $CH_4$ fluxes have a strong, positive correlation with the $CO_2$ uptake (Bellisario et al., 1999; Christensen et al., 2000; Rinne et al., 2018), since higher primary productivity leads to a higher input of substrate. Of the vegetation components, deep-rooting aerenchymatous species such as sedges (Cyperaceae) and aerenchymatous herbs are especially important (Leppälä et al., 2011; Ward et al., 2013). In sedge-dominated wetlands, most of the $CH_4$ is released through vascular plants (Kelker and Chanton, 1997; Ding et al., 2004; Ström et al., 2005), thus bypassing the aerobic peat layer where $CH_4$ oxidation takes place. On the other hand, oxygen transport through the aerenchyma to the rhizosphere may inhibit $CH_4$ production (Whalen and Reeburgh, 2000; Fritz et al., 2011) and stimulate $CH_4$ oxidation (King, 1994; Popp et al., 2000). The net effect of the presence of aerenchymatous species on $CH_4$ fluxes is positive in most cases (Bellisario et al., 1999; Greenup et al., 2000; Rinnan et al., 2003; Couwenberg and Fritz, 2012; Ward et al., 2013), although opposite results have also been reported (Roura-Carol and Freeman, 1999; Strack et al., 2006). Although the influence of the non-aerenchymatous species on the fluxes has been studied relatively little, Gray et al. (2013) showed that plant functional groups based on more complex traits than those related to aerenchyma were good proxies of $CH_4$ flux. In open boreal peatlands, the most abundant non-aerenchymatous vascular plant functional group is dwarf shrubs, that are generally shallow rooted (Korrensalo et al., 2018a) and have a negligible $CH_4$ transport capacity (Shannon et al., 1996; Garnet et al., 2005) compared to deep-rooting aerenchymatous species. In plant removal experiments, the presence of shrubs has been shown to decrease $CH_4$ fluxes (Ward et al., 2013; Robroek et al., 2015). Recently, trees have been shown to transport significant amounts of $CH_4$ from soil in certain ecosystems, but so far not in forested boreal peatlands (Covey and Megonigal, 2019). *Sphagnum* mosses, in turn, have an impact on $CH_4$ oxidation as they host partly endophytic methanotrophs in the water-filled, hyaline cells of their leaves and stem (Raghoebarsing et al., 2005; Larmola et al., 2010; Putkinen et al., 2012).

Water level regulates the volume ratio of the aerobic and anaerobic peat and, consequently, the extent of the $CH_4$ production and oxidation zones. Therefore, a positive correlation between the water level and $CH_4$ fluxes has been reported in numerous studies (Moore and Roulet, 1993; Laine et al., 2007a; Pearson et al., 2015; Turetsky et al., 2014; Chimner et al., 2017). However, the relationship between the water level and $CH_4$ fluxes is complex due to the vegetation – water level interaction. Because the plant communities in the wettest habitats are often associated with the sparsest vascular plant cover and lowest productivity (Waddington and Roulet, 2000; Laine et al., 2007b; Riutta et al., 2007b), less substrate for $CH_4$ production is available in those communities. In the dry end of the water level gradient, fewer roots reach the anaerobic layer of the peat (Waddington et al., 1996; Kutzbach et al., 2004). Hence, $CH_4$ fluxes

may also show a unimodal relationship to water level (Strack et al., 2004; Brown et al., 2014) or no relationship at all (Rask et al., 2002; Korrensalo et al., 2018b).

In this study, we aim to disentangle the intertwined relationships among water level, vegetation and fen $CH_4$ fluxes. We test the role assumed for different plant functional groups based on earlier literature and quantify how these roles are modulated by changing water level. Our objective is to quantify the contribution of the different components of fen plant community, namely sedges, dwarf shrubs, *Sphagnum* mosses, and the underlying peat, to the $CH_4$ fluxes under wet and dry conditions. $CH_4$To achieve this, we applied removal treatments of plant functional groups both under natural and experimentally lowered water level in a factorial study design. We hypothesized that aerenchymatous plant species enhance $CH_4$ fluxes and that this effect would be less pronounced under lowered water level as a smaller proportion of the roots would extend to the anaerobic peat layer. Further, we hypothesized *Sphagnum* mosses and dwarf shrubs to reduce $CH_4$ fluxes.

## 2. Materials and methods

### 2.1. Study site

The study was carried out at Lakkasuo peatland complex, an eccentric raised bog with minerotrophic laggs situated on the  southern boreal vegetation zone (Ahti et al., 1968) in Southern Finland (61°47' N; 24°18' E). Annual precipitation in the region totals 710 mm, of which about a third falls as snow. The average temperatures for January and July are -8.9 and 15.3 °C, respectively (Juupajoki-Hyytiälä weather station, Drebs et al. 2002).

The study site was situated on a nutrient-poor, oligotrophic, treeless fen part of the peatland complex. Surface topography in the site is uniform, mostly lawn. The pH of the surface peat at the site was 4.9 (Juottonen et al., 2005). Field layer is dominated by sedges and dwarf shrubs. The most abundant sedge species is *Carex lasiocarpa* Ehrh. (% cover in 2001 3.4 ± 3.9, mean ± standard deviation of 40 inventory plots), and other common sedge species are *Eriophorum vaginatum* L. (0.9 ± 1.8) and *Trichophorum cespitosum* (L.) Hartm. (0.5 ± 2.4). The most abundant shrubs are the deciduous *Betula nana* L. (4.0 ± 4.2) and ericaceous *Andromeda polifolia* L. (6.6 ± 5.7) and *Vaccinium oxycoccos* L. (4.9 ± 4.2). Note that due to the erect growth form of sedges, their % cover is lower than that of shrubs, although their leaf area is higher; see Table 1 and Fig. 2. The moss layer forms a continuous carpet dominated by *Sphagnum papillosum* Lindb. (40.1 ± 31.3) and the species of *S. recurvum* complex (*S. fallax* (Klinggr.) Klinggr., *S. flexuosum* Dozy & Molk and *S. angustifolium* (C.E.O.Jensen ex Russow) C.E.O.Jensen) (together 32.7 ± 24.0). The vegetation inventory and variation conducted at the site is described in detail in Kokkonen et al., (2019).

### 2.2. Experimental design

The study was carried out during four growing seasons from 2001 to 2004. The first season of the study, 2001, served as a calibration year without the WLD treatment, which was implemented in April 2002. The study site was divided into two subsites approximately 20 m apart, namely the wet and the drier water level drawdown (WLD) subsite, by surrounding the WLD subsite with a shallow ditch that lowered the water level by an average $17 \pm 1.6$ cm ($22 \pm 3.0$ cm in 2002, $12 \pm 3.4$ cm in 2003 and $16 \pm 1.9$ cm in 2004). The shallow ditch was located approximately 10 m from the wet subsite and drained to a larger, old ditch.

We studied the contribution of the ecosystem components to the net $CH_4$ fluxes in wet and dry conditions by means of plant removal treatments. In the site, we established permanent sample plots of 56 cm $\times$ 56 cm consisting of:

- peat, *Sphagnum* mosses, sedges and dwarf shrubs (PSCD, intact vegetation, n = 8 in the wet subsite and n = 8 in WLD subsite)
- peat, *Sphagnum* mosses and sedges (PSC, dwarf shrubs removed, n = 5 + 4)
- peat and *Sphagnum* mosses (PS, sedges and shrubs removed, n = 3 + 3)
- peat (P, all vegetation removed, n = 4 + 4).

The plant removal treatment plots (PSC, PS and P) were established April 2002. In the plant removal treatment plots vascular plants were cut with scissors to the level of the moss (PS plots) or peat (P plots) surface and their above-ground litter was removed. In the P plots the top 1.5 cm of the *Sphagnum* moss carpet was cut off with scissors. All emerging regrowth was clipped off once a week as necessary. Over the course of the study, progressively less clipping was needed, hardly any in 2004. Prior to $CH_4$ flux measurements, sedge stubble in P and PS plots was treated with paraffin wax to seal the aerenchymatous pathway of $CH_4$.

### 2.3. Measurements

$CH_4$ fluxes were measured using the closed chamber method. A stainless steel collar ($56 \times 56 \times 30$ cm, length x width x height) was permanently inserted into each sample plot prior to the start of the study. The collars had a water groove to allow chamber placement and air-tight sealing during the measurement. For the flux measurements, an aluminium chamber of $60 \times 60 \times 30$ cm was placed on the water groove of the collar. After the chamber placement, a vent on the chamber roof that ensured pressure equilibration was sealed with a septum plug. A battery-operated fan circulated the air inside the chamber. A 40-ml air sample was drawn into a polypropylene syringe at 5, 15, 25 and 35 minutes after closure. The samples were stored at $+ 4°C$ before analysis, which was carried out within 36 hours. Samples were analyzed with a HP-5710A gas chromatograph (GC) from 2001 to 2003 and with a HP-5890A GC in 2004. Both GCs were equipped with a 1-ml loop, $6 \times 1/8$" packed column (Hayesep Q in HP-5710A; Poropak Q in HP-5890A) and flame ionization detector. The carrier gas was helium with a flow rate of 30 mL min$^{-1}$. Column and detector temperatures were 40°C and 300°C, respectively. The precision of the analysis was $\pm 0.16\%$, determined as the coefficient of variation of the replicate samples.

To relate the fluxes to prevailing environmental conditions, peat temperatures at 5, 10, 20 and 30 cm below the moss surface and water level in a perforated tube adjacent to each plot were measured during the flux measurements. Air and peat temperatures and precipitation were also continuously recorded in the weather station at the site. Green leaf area index (LAI) of each vascular plant species in each plot was determined with the method of Wilson et al. (2007) from April until November, as a product of the total number of leaves (counted monthly) and the average leaf size of marked individuals (measured every two weeks). Species-specific Gaussian curves (Wilson et al. 2007) were fitted to the observations to describe the continuous development of LAI throughout the season. LAI of different species were summed up to sedge, dwarf-shrub and total LAI ($LAI_C$, $LAI_D$ and $LAI_T$, respectively). Moss cover at each plot was visually estimated annually.

In addition to $CH_4$ exchange, $CO_2$ exchange was measured in the study site. The methods and results are reported elsewhere (Riutta et al., 2007a) in more detail, but some $CO_2$ exchange estimates are used here to study the relationship between the $CO_2$ and $CH_4$ fluxes. In summary, net ecosystem $CO_2$ exchange (NEE) was measured weekly / biweekly by employing the closed chamber technique in the same plots and during the same period as the $CH_4$ fluxes. Measurements were carried out in both light and dark, which enabled the partitioning of the fluxes into gross photosynthesis and ecosystem respiration. We constructed nonlinear regression models for photosynthesis and respiration, with water level, temperature and LAI as explanatory factors, separately for each vegetation treatment, to reconstruct the fluxes for the whole growing season.

### 2.4. Data analyses

$CH_4$ flux was calculated as the linear change in $CH_4$ concentration as a function of time by fitting a least-squares regression line. Of the 1300 measurements, <0.5% were rejected due to clear errors, such as leakage or problems in the GC analysis, and 2% were classified as episodic fluxes.

To reconstruct seasonal (June-September) estimates for each sample plot, the biweekly measured fluxes were linearly interpolated between measurement days and the obtained daily values were integrated. In the interpolation, rejected values and episodic fluxes were replaced with the median flux of the corresponding vegetation and water level treatment on the same measurement day. The impact of the episodic fluxes on the seasonal flux was taken into account by using the episodic values as the $CH_4$ flux estimates of the day they were measured. The reconstructed seasonal fluxes at the wet and WLD subsites were converted to $CO_2$ equivalent according to Myhre et al., (2013).

We used linear mixed effect models to test the impact of the plant removal treatments and the WLD treatment on WL, LAI and daily measured $CH_4$ flux. First, we tested the differences in WL, $LAI_C$, $LAI_D$, $LAI_T$ and $CH_4$ flux between the wet and WLD subsites before the WLD treatment was applied (year 2001) and over the years after the WLD treatment (2002-2004), with WLD treatment, year and their interaction as potential fixed predictors. This model included only the plots with intact vegetation (PSCD). The wet subsite in 2001 was the constant against which WLD and other years were compared.

Therefore, the difference between the wet and WLD treatment in the model describes the pre-treatment difference among the two subsites in calibration year 2001, and the interaction between WLD treatment and years 2002-2004 describe the impact of WLD after the treatment.

Second, we tested the impact of plant removal on $CH_4$ flux over the years and the interaction of the plant removal treatments with the WLD treatment with data from years 2002-2004 (no plant removal treatments in 2001). For each year separately, we fitted a model with plant removal treatments, WLD treatment and the interaction between them as potential fixed predictors.

Third, we tested the response of $CH_4$ flux to leaf area and environmental variables by extending the model fitted to the data of year 2004, that had the maximum amount of time for stabilization after the treatments. In addition to plant removal and WLD treatments, potential fixed predictors were $LAI_C$, $LAI_D$, cover of *Sphagnum* mosses, measured WL, temperature in the chamber and peat temperature at the depths of 5, 10 20 and 30 cm (T5, T10, T20 and T30) as well as the potential interactions among these parameters. Potential new predictors were sequentially added and after each addition the significance of all predictors were tested. We reported separately both models for year 2004; one including plant removal and WLD treatments as fixed predictors for $CH_4$ flux and another including the response of $CH_4$ flux to leaf area/cover of plant groups and environmental variables.

In each case, a conditional F-test was used to test if the full model with all fixed predictors and their interactions was significantly better ($p<0.05$) than a simpler model. Plot and date were included as crossed random effects. Resulting models are reported in Table 2. The models were fitted using the function lmer of package lme4 (Bates et al., 2015) of RStudio version 1.1.383.

### 3. Results

#### 3.1. Impact of the water level drawdown

The pre-treatment water level did not differ between the wet and WLD subsites ($p=0.174$, comparison between wet and WLD treatment during the calibration year 2001) (Fig. 1a, Table 1). Following the drainage in April 2002, the water level was significantly lower in the WLD subsite ($p < 0.001$, interaction between WLD and year 2002). The WLD treatment lowered the water level by approximately 17 cm, except in July and August 2003 when a severe drought lowered the water level below the ditch, resulting in similar water levels in wet and WLD subsites. In the wet subsite, the water level during the years 2001 and 2004 was similar to the long-term average of the site (Laine 2004), approximately 5 to 10 cm below the moss surface (Table 1) (Laine et al. 2004). During July and August 2002 and 2003, however, the water level was lower than the long-term average. More information on the weather conditions during the study is given in Riutta et al. (2007b).

Prior to the drainage, vegetation composition in the plots with intact vegetation (PSCD) was similar in both subsites (Table 1, Fig. 1b). In the mixed effects model, $LAI_C$, $LAI_D$ and $LAI_T$ did not differ between wet and WLD subsites in year 2001 (p-values 0.996, 0.656 and 0.878, respectively). In 2001 the peak

season average $LAI_T$ was approximately 1.0 $m^2$ $m^{-2}$, of which sedges composed 70%. The mean *Sphagnum* cover was 80%. By the third year since WLD 2004 $LAI_C$ had decreased (p<0.001) and $LAI_D$ increased (p<0.001) in the WLD subsite, resulting in an overall decrease in $LAI_T$ (p=0.007) (Table 1, Fig. 1b).

In the PSCD plots, the pre-treatment $CH_4$ fluxes did not differ between the wet and WLD subsite (p=0.654) (Fig. 1c). After the treatment, in 2002-2004, fluxes were significantly lower in the WLD than in the wet subsite (p<0.001 for all years). During the three-year WLD treatment, the mean flux was approximately 51 and 7.0 mg $CH_4$ $m^{-2}$ $d^{-1}$ in the wet and WLD subsites, respectively. Converted to $CO_2$ equivalents, the seasonal reconstructed fluxes in the wet and WLD subsites in 2002-2004 were 236 and 32 g $CO_2$-eq $m^{-2}$ growing season$^{-1}$, respectively.

### 3.2. Impact of the plant removal treatments

Plant removal treatments did not lead to major changes in vegetation composition beyond the clipped target groups. Vascular plant removal did not affect the *Sphagnum* moss cover, and the removal of dwarf shrubs did not change the LAI of sedges. $LAI_C$ was similar in PSC and PSCD plots (data for 2004 shown in Table 1) during all years in the wet subsite and during 2003 and 2004 in the WLD subsite (all p-values >0.05). $LAI_C$ was higher in the PSC plots than in the PSCD plots in the WLD subsite in 2002 (p=0.016).

During the first season of the removal treatments (2002) in the wet subsite, $CH_4$ fluxes were higher in the plant removal plots (P, PS and PSC) than in the intact plots (PSCD), in some cases almost triple (p<0.05 for all treatments, Fig. 2, upper panels). The fluxes in the plant removal treatment plots also showed a stronger seasonal pattern and larger spatial variation. After the first year of removal treatments the fluxes of the P, PS and PSC plots decreased, and in 2003 P plots had a similar $CH_4$ flux than the intact plots (p=0.908), while PS and PSC plots still had a higher flux than PSCD plots (p=0.033 and p=0.005, respectively).

By the third year of the plant removal treatments (2004), the fluxes in all treatments showed a seasonal pattern similar to that of the intact plots. Bare peat plots had lower fluxes than the intact PSCD plots (p<0.001). Fluxes of the PSC plots (shrubs removed) were marginally significantly higher (p=0.060) than those of the PSCD plots (shrubs present). In WLD conditions, the fluxes in the plant removal plots (P, PS and PSC) were mostly lower than the fluxes in the intact PSCD plots during all three vegetation treatment years (Fig. 2, lower panel), but the differences were not significant (Table 2b). WLD and plant removal treatments had a significant interaction: in 2004 WLD lowered the fluxes more in PSCD plots than in the P plots and more in PSC plots than in the P and PS plots (p<0.05 for the interaction terms). Seasonal fluxes visualize the patterns tested with the nonlinear mixed effect models: in the WLD subsite fluxes were lower than in the wet subsite in all plant removal treatments (Fig. 3b). In wet conditions, the seasonal flux of the P and PS plots was lower than that of the PSCD and PSC plots in which vascular plants were present (Fig. 3a). Taking the fluxes from bare peat plots as a baseline, the presence of vegetation enhanced the fluxes. Compared with the situation of sedges and *Sphagna* present (PSC), the

presence of shrubs (PSCD) seemed to slightly attenuate the fluxes (Fig. 3b, c). In WLD conditions, the differences between plant removal treatments were negligible. The differences between the plant removal treatments can be used as an estimate of the contribution of each plant group to the total flux, although due to the propagation of the errors, uncertainty in these estimates is large. In normal hydrological conditions, plant-mediated flux accounted for 68% ± 23% (comparison of P and PSCD plots) or 78% ± 17% (comparison of P and PSC plots) of the total growing season flux, of which *Sphagnum* mosses and sedges accounted for approximately ¼ and ¾, respectively (Fig. 3c).

The seasonal $CH_4$ fluxes displayed a clear positive, exponential relationship with the seasonal net $CO_2$ flux (Fig. 4). The relationship was similar among the plant removal treatments in wet and dry conditions. However, the plots with intact vegetation (PSCD) were an exception; they had lower $CH_4$ fluxes than

could have been expected based on their net $CO_2$ flux, pointing towards the potential suppressing effect of shrubs on $CH_4$ emissions.

### 3.3. Response of $CH_4$ flux to environmental variables and interaction with leaf area

The best predictors of the $CH_4$ flux in the extended model for the year 2004 were the categorical WLD treatment (which was a better predictor than the measured WL), T20 (best out of the measured temperatures), and $LAI_C$ (which was a better predictor than the categorical vegetation removal treatment). The abundance of the other plant functional groups, $LAI_D$, or *Sphagnum* cover did not have a significant effect on the fluxes. $CH_4$ flux was increased by $LAI_C$ and T20 in wet conditions (Table 2c). In the WLD conditions, however, neither $LAI_C$ nor T20 had any impact on the fluxes (coefficient estimates for $LAI_C$*WLD1 and T20*WLD1 cancel out the coefficient estimates for $LAI_C$ and T20 in wet conditions;

Table 2c). The positive coefficient of the WLD treatment seemingly indicated a larger flux at the WLD treatment site compared with the wet site, when $LAI_C$ and T20 both equal zero; however, the measured minimum T20 during the growing season 2004 was 6.1°C, and the model was not intended for any extrapolation. The predicted $CH_4$ flux in the WLD treatment was similar to or lower than the flux in the wet treatment in the observed T20 and $LAI_C$ range.**4. Discussion**

### 4.1. The effect of the plant types and substrate on the $CH_4$ fluxes in natural water level conditions

In line with the previous studies, the plant removal treatments of this study indicated that sedges were the most important plant group in regulating $CH_4$ fluxes. In other sedge-dominated sites, plant-mediated flux has accounted for 75 to 97% of the total flux (Schimel, 1995; Kelker and Chanton, 1997; Ström et al., 2005; Sun et al., 2012; Noyce et al., 2014) and plant removal experiments have shown that of different

plant functional types, removal of graminoids cause the largest decrease on $CH_4$ production and flux (Ward et al., 2013; Robroek et al., 2015). Compared with the bare peat surfaces, the presence of *Sphagnum* mosses seemed to have a slight, although not statistically significant, enhancing effect on the $CH_4$ fluxes, similarly to the results of Roura-Carol and Freeman (1999). King et al. (1998) found the presence of mosses to have a slightly attenuating effect on the fluxes, while Greenup and others (2000) did not find significant differences in fluxes after *Sphagnum* removal. Based on this, the $CH_4$ oxidation

by the loosely symbiotic methanotrophs within *Sphagnum* mosses (Raghoebarsing et al., 2005; Larmola et al., 2010; Putkinen et al., 2012) seems to play a minor role in $CH_4$ dynamics in our site.

Similarly to Ward et al. (2013), we found that the presence of shrubs seemed to have a slightly attenuating effect on the fluxes under natural water level conditions. Robroek et al. (2015) made a similar finding with potential $CH_4$ production. In contrast, an aerenchymatous shrub, *Myrica gale*, supported similar potential $CH_4$ production than a sedge, *Carex aquatilis*, and did not suppress $CH_4$ flux (Strack et al., 2017). Furthermore, in line with the attenuating effect of shrubs, the $CH_4$ flux: NEE ratio was lower in the plots with intact vegetation (PSCD, shrubs present) than in the other vegetation treatments. Mechanisms behind that might relate to impact of shrubs on soil chemistry, microbial community or the biomass allocation of sedges. Shrub litter has higher lignin and leaf dry matter content than sedges, which both are related to lower methanogenesis (Yavitt et al., 2019). Shrub removal has been observed to result in higher dissolved organic C and N and lower C:N ratio (Ward et al. 2013) as well as higher fungal biomass (Robroek et al. 2015). A study on the competitive ability and biomass allocation of a wetland grass, *Molinia caerulea*, revealed that *M. caerulea* allocated more biomass to the roots when it did not face competition by shrubs (Aerts et al., 1991). Similarly, in our study, sedges in the plots where shrubs were removed may have allocated more biomass to the roots than the sedges growing in the sedge and shrub mixture. As a result, methanogenic microbes may have benefited from the higher substrate availability in the shrub removal plots (PSC). $CH_4$ production has a negative relationship and $CH_4$ oxidation has a positive relationship with the concentration of certain woody lignin compounds in peat pore water (Yavitt, 2000). In our study, this may be the reason behind the lower fluxes in the presence of the arboreals. The results concerning the attenuating effect of shrubs on $CH_4$ fluxes are, however, only indicative and further, process orientated research is needed.

### 4.2. Delay in the plant removal treatment effect

We observed a considerable disturbance in the fluxes following the plant removal treatments. In other clipping studies in which the shoots were cut above the water level clipping either increased the $CH_4$ flux during the first growing season after clipping (Schimel, 1995), had no effect (Kelker and Chanton, 1997; Greenup et al., 2000) or decreased the flux (Waddington et al., 1996; Rinnan et al., 2003). Thus, we assumed that the higher fluxes at the clipped plots during the first two years after the vegetation removal treatments were mainly caused by treatment artefacts. The removal of the above-ground parts of vascular plants led to the gradual death of the below-ground parts, creation of unnatural amount of new root necromass and, thereby, a peak in the amount of available substrate. Methanogenesis in the study site may have been substrate limited (Bergman et al., 1998; Rinne et al., 2007), which could explain the initially high fluxes in the plant removal plots. The mass loss of *Carex* roots and rhizomes is only 10 to 45% during the first 12 months of decomposition, although the litter quality deteriorates (Scheffer and Aerts, 2000). However, after two years the mass loss can be as much as 75% of the original mass (Thormann et al., 2001), which gives more confidence in the results of the third year of the plant removal treatments. Thus, we used the third year of the plant removal treatments to quantify the contribution of the vegetation components to the fluxes and the response of fluxes to environmental conditions. King

and others (1998) likewise reported the effects of the plant removal two years after the treatment began. Shrub litter, especially below-ground litter, decomposes slower than sedge litter (Moore et al., 2007), due to the high lignin content (Yavitt et al., 2019). On the other hand, the majority of dwarf shrub roots grow in the uppermost 20 cm peat layer, while sedge roots extend deeper (Korrensalo et al., 2018a; Mäkiranta et al. 2018), causing a larger proportion of dwarf shrub roots to decompose in oxic conditions, thus counteracting the differences in litter quality. Even two years after the start of the vegetation removal treatments, some shrub roots still probably remained. However, they were mostly located above the $CH_4$ production zone.

### 4.3. Water level regulates the role of the vegetation

Experimental water level drawdown has been used to mimic climate change impact on northern peatland $CH_4$ fluxes in mesocosm (Freeman et al., 1992; Blodau et al., 2004; Dinsmore et al., 2009) and in the field studies ranging from bogs to rich fens (Laine et al., 2007a; Strack and Waddington, 2007; Turetsky et al., 2008; Ballantyne et al., 2014; Munir and Strack, 2014; Pearson et al., 2015; Peltoniemi et al., 2016; Chimner et al., 2017; Olefeldt et al., 2017). In line with our results, all these studies report some level of decrease in $CH_4$ flux due to WLD ranging from 3 to ~20cm. Together with temperature and vegetation, water level is a major regulator of $CH_4$ flux (Lai, 2009; Turetsky et al., 2014). However, the mechanistic understanding of this process is still limited. While Strack et al. (2004) found only small differences in the $CH_4$ production and consumption potentials between control and WLD sites, and thus attributed the the decrease in fluxes mainly to the change in the volume ratio of the anaerobic and aerobic zones, Yrjälä et al. (2011) and Peltoniemi et al. (2016) found that WLD had a stronger impact on emissions through decreasing $CH_4$ production, than through increasing oxidation,

In this study, the presence or absence of the plant types or $LAI_C$ had no effect on the $CH_4$ flux in the WLD conditions. This supports the findings by Waddington et al. (1996) as well as Strack et al. (2006) that the impact of the vegetation on the fluxes is strongly dependent on the water level conditions. $CH_4$ flux also responded to peat temperature only in wet conditions. A similar result with water level and temperature response has been previously reported by Moosavi et al., (1996). Our results showed that water level acts as a switch; it turns $CH_4$ flux on and off, after which temperature and vegetation regulate the flux magnitude. This result is further emphasized by the response model, where WLD treatment including change in the ecosystem following new WT regime rather than seasonally varying WL was a better predictor for $CH_4$ fluxes. In conclusion, vegetation is a major controlling factor of the peatland $CH_4$ dynamics, but only in wet conditions.

### 5. Conclusions

Vegetation, sedges in particular, regulates the level of fen $CH_4$ fluxes in normal hydrological conditions, but this vegetation control is strongly dependent on the water level regime. In water level drawdown conditions $CH_4$ fluxes are significantly lowered, practically to zero, and vegetation composition has no influence on the fluxes. The results are relevant for assessing the response of fen peatlands to changing

climatic conditions, as water level drawdown and the consequent vegetation changes are the major projected impacts of climate change on northern peatlands.

**Competing interests**

The authors declare that they have no conflict of interest.

**Data availability**

The data associated with the manuscript will be published in PANGAEA repository upon publication. On request, the data can also be obtained from the corresponding author.

**Acknowledgements**

We thank Jouni Meronen for technical support and the field team for assistance in the study site. Meeri Pearson kindly revised the language. Financial support was received from the Academy of Finland (SA287039, SA50707, SA201623, SA202424), Jenny and Antti Wihuri Foundation, Kone Foundation, Faculty of Science and Forestry of University of Eastern Finland and the Graduate School in Forest Sciences of University of Helsinki.

**Author contribution**

The study was designed and the field experiment established by EST, TR and JL. Field work, flux calculation and flux reconstruction was conducted by TR. Statistical analysis was done by AK. TR wrote the first version of the manuscript, which was further processed by all other authors.

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

**Table 1. Growing season average (standard deviation) water level (WL) relative to moss surface (unit is cm), negative values indicating water level below the surface, growing season peak LAI of sedges (LAI$_C$) and dwarf shrubs (LAI$_D$), and projection cover of *Sphagnum* mosses (Spha) (units are m$^2$ m$^{-2}$) in different plant removal treatments in wet and WL drawdown subsites. Year 2001 was a calibration year without the WL drawdown and plant removal treatments, which were implemented in 2002. Vegetation treatments: PSCD - plots with intact vegetation, consisting of peat, *Sphagnum* mosses, sedges and shrubs;  PSC - plots consisting of peat, *Sphagnum* mosses and sedges (shrubs removed); PS - plots consisting of peat and *Sphagnum* mosses (shrubs and sedges removed), P- plots consisting of bare peat (all vegetation removed).**

| Year | Vegetation | Wet WL | Wet LAI$_C$ | Wet LAI$_D$ | Wet Spha | WL drawdown WL | WL drawdown LAI$_C$ | WL drawdown LAI$_D$ | WL drawdown Spha |
|------|-----------|----|------|------|------|----|------|------|------|
| 2001 | PSCD | -7 (4) | 0.7 (0.3) | 0.2 (0.1) | 0.8 (0.2) | -5 (3) | 0.7 (0.3) | 0.3 (0.2) | 0.8 (0.1) |
| 2004 | PSCD | -10 (4) | 0.6 (0.2) | 0.3 (0.1) | 0.9 (0.1) | -24 (6) | 0.3 (0.1) | 0.3 (0.1) | 0.6 (0.2) |
|      | PSC | -10 (5) | 0.7 (0.5) | | 0.7 (0.2) | -29 (7) | 0.8 (0.3) | | 0.7 (0.3) |
|      | PS | -11 (3) | | | 0.8 (0.2) | -26 (7) | | | 0.7 (0.2) |
|      | P | -7 (4) | | | | -21 (8) | | | |

**Table 2. Parameter estimates of the linear mixed-effects model describing (a) the differences, water level (WL, cm), total, sedge and dwarf-shrub leaf area index (LAI$_T$, LAI$_C$ and LAI$_D$), and CH$_4$ flux between wet (WLD0) and water level drawdown (WLD1) subsites and years before (2001) and after (2002-2004) the WLD treatment in plots without vegetation removal, (b) the differences in CH$_4$ flux between the vegetation removal treatments in years 2002-2004 and (c) the response of CH$_4$ flux in year 2004 to leaf area and environmental variables. Vegetation treatments: PSCD – intact vegetation, PSC - plots consisting of peat, *Sphagnum* mosses and sedges (shrubs removed); PS - plots consisting of peat and *Sphagnum* (sedges and shrubs removed); P - plots consisting of bare peat (all vegetation removed).**

| (a) | WT | | | LAI$_T$ | | | LAI$_C$ | | | LAI$_D$ | | | CH$_4$ flux | | |
| **Fixed part** | Coeff. | SE | P-value | Coeff. | SE | P-value | Coeff. | SE | P-value | Coeff. | SE | P-value | Coeff. | SE | P-value |
|---|---|---|---|---|---|---|---|---|---|---|---|---|---|---|---|
| Constant (WLD0, Year 2001) | -6.9 | 2.8 | 0.018 | 0.6 | 0.1 | <0.001 | 0.5 | 0.1 | <0.001 | 0.2 | 0.04 | <0.001 | 3.1 | 0.5 | <0.001 |
| WLD1 | 1.6 | 1.1 | 0.174 | 0.02 | 0.1 | 0.878 | 0.0006 | 0.1 | 0.996 | 0.02 | 0.04 | 0.656 | 0.2 | 0.3 | 0.654 |
| Year 2002 | -8.2 | 3.6 | 0.030 | 0.2 | 0.1 | 0.118 | 0.2 | 0.1 | 0.031 | -0.03 | 0.03 | 0.366 | -0.3 | 0.6 | 0.571 |
| Year 2003 | -14.2 | 3.7 | <0.001 | -0.02 | 0.1 | 0.879 | -0.007 | 0.1 | 0.952 | -0.02 | 0.03 | 0.642 | -2.0 | 0.6 | 0.002 |
| Year 2004 | -2.1 | 3.5 | 0.550 | -0.1 | 0.1 | 0.364 | -0.1 | 0.1 | 0.214 | 0.01 | 0.03 | 0.648 | -1.3 | 0.6 | 0.034 |
| WLD1*Year 2002 | -16.6 | 0.7 | <0.001 | -0.06 | 0.05 | 0.173 | -0.1 | 0.04 | 0.029 | 0.03 | 0.01 | 0.006 | -2.3 | 0.3 | <0.001 |
| WLD1*Year 2003 | -11.3 | 0.7 | <0.001 | -0.06 | 0.05 | 0.192 | -0.1 | 0.05 | 0.034 | 0.04 | 0.01 | 0.006 | -1.1 | 0.3 | <0.001 |
| WLD1*Year 2004 | -16.3 | 0.7 | <0.001 | -0.1 | 0.05 | 0.007 | -0.2 | 0.05 | <0.001 | 0.05 | 0.01 | <0.001 | -2.1 | 0.3 | <0.001 |
| | | | | | | | | | | | | | | | |
| **Random part** | | | | | | | | | | | | | | | |
| SD (Measurement day) | 7.5 | | | 0.28 | | | 0.22 | | | 0.06 | | | 1.2 | | |
| SD (Plot code) | 2.0 | | | 0.25 | | | 0.21 | | | 0.09 | | | 0.5 | | |
| Residual SD | 2.9 | | | 0.19 | | | 0.19 | | | 0.05 | | | 1.1 | | |

| (b) | 2002 | | | 2003 | | | 2004 | | |
| **Fixed part** | Coeff. | SE | P-value | Coeff. | SE | P-value | Coeff. | SE | P-value |
|---|---|---|---|---|---|---|---|---|---|
| Constant (PSCD, WLD0) | 3.8 | 1.1 | <0.001 | 1.4 | 0.3 | <0.001 | 1.9 | 0.3 | <0.001 |
| P | 2.8 | 1.3 | 0.040 | 0.1 | 0.5 | 0.908 | -1.2 | 0.3 | <0.001 |
| PS | 6.0 | 1.6 | 0.001 | 1.3 | 0.6 | 0.033 | -0.4 | 0.4 | 0.373 |
| PSC | 7.2 | 1.5 | <0.001 | 1.5 | 0.5 | 0.005 | 0.7 | 0.4 | 0.060 |
| WLD1 | -2.2 | 1.0 | 0.036 | -1.0 | 0.4 | 0.022 | -1.9 | 0.3 | <0.001 |
| P*WLD1 | -4.4 | 1.8 | 0.020 | -0.4 | 0.7 | 0.622 | 1.2 | 0.4 | 0.008 |
| PS*WLD1 | -8.2 | 1.9 | <0.001 | -1.6 | 0.8 | 0.049 | 0.6 | 0.5 | 0.256 |
| PSC*WLD1 | -9.0 | 1.6 | <0.001 | -1.9 | 0.7 | 0.009 | -0.5 | 0.4 | 0.224 |
| | | | | | | | | | |
| **Random part** | | | | | | | | | |
| SD (Measurement day) | 1.6 | | | 0.7 | | | 0.9 | | |
| SD (Plot code) | 3.0 | | | 0.7 | | | 0.4 | | |
| Residual SD | 3.8 | | | 1.3 | | | 0.9 | | |

| (c) | | | |
| **Fixed part** | Coeff. | SE | P-value |
|---|---|---|---|
| Constant (WLD0) | -1.8 | 0.4 | <0.001 |
| LAI$_C$ | 2.5 | 0.3 | <0.001 |
| T20 | 0.2 | 0.03 | <0.001 |
| WLD1 | 1.6 | 0.4 | <0.001 |
| LAI$_C$*WLD1 | -2.5 | 0.5 | <0.001 |
| T20*WLD1 | -0.2 | 0.03 | <0.001 |
| | | | |
| **Random part** | | | |
| SD (Measurement day) | 0.5 | | |
| SD (Plot code) | 0.4 | | |
| Residual SD | 0.8 | | |

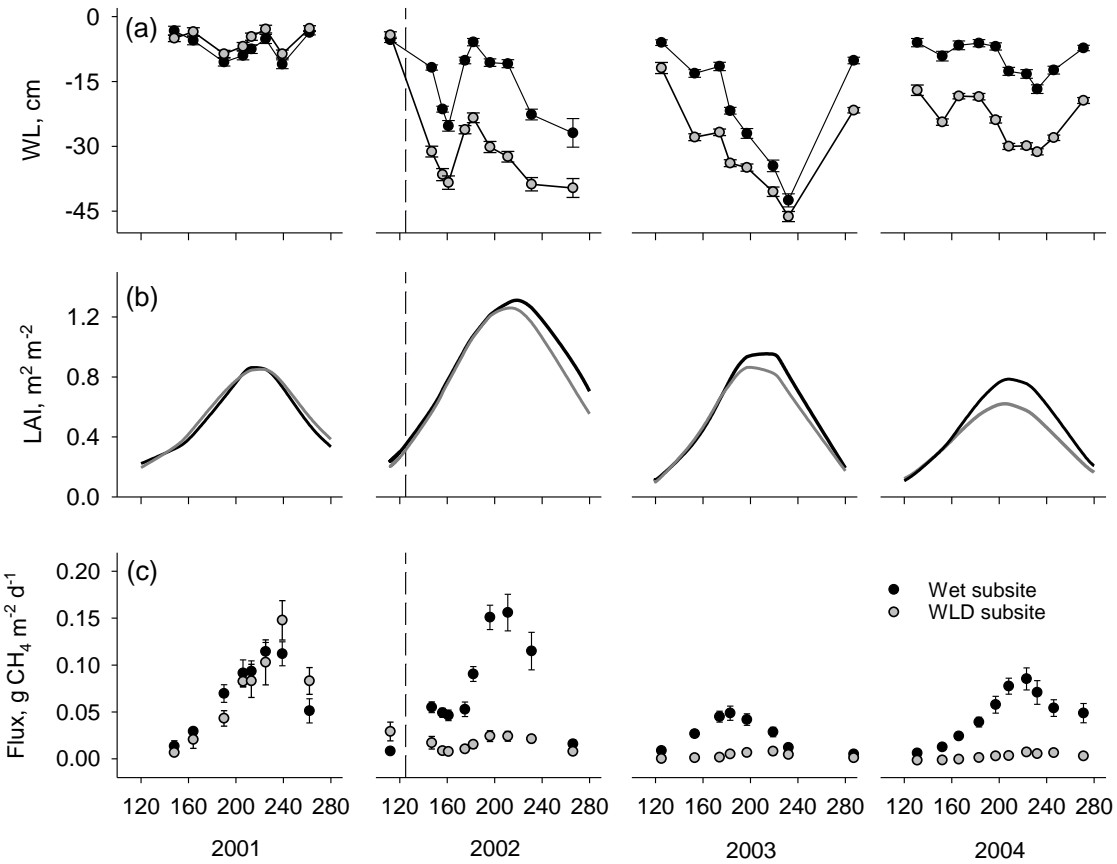

**Figure 1. Mean a) water level (WL), b) leaf area index (LAI), and c) CH₄ flux in plots with intact vegetation in wet and water level drawdown (WLD) subsites. Error bars are standard errors of the mean. Units on the x-axis give the day of year. The start of the water level drawdown treatment is indicated with the vertical dashed line in 2002. Water level is negative when it is below the moss surface. Positive CH₄ fluxes indicate emission to the atmosphere.**

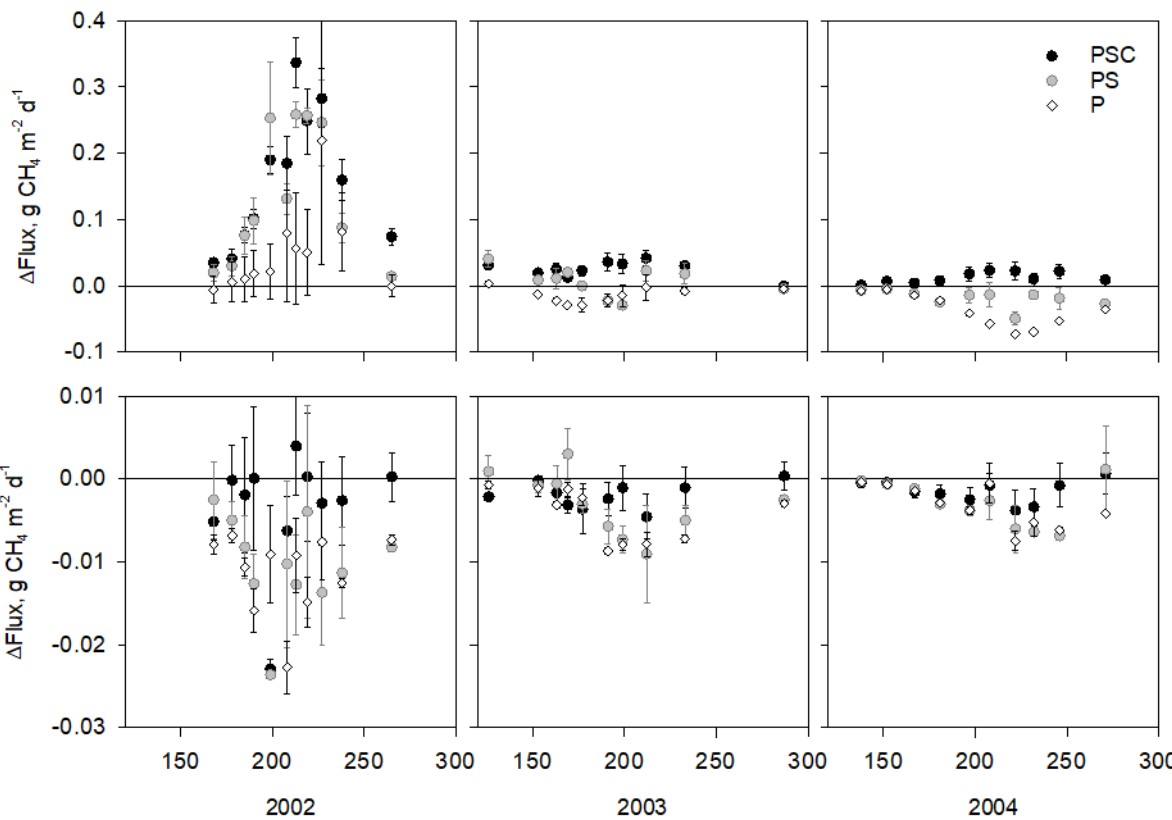

**Figure 2. Difference of the measured CH₄ fluxes in plots with plant removal treatments and the mean flux in the plots with intact vegetation on each measurement day in control subsite (upper panels) and water level drawdown subsite (lower panels). Positive values indicate that fluxes in the plant removal treatment plots are higher than in the intact plots. Units on the x-axis give the day of year. Note the difference scales of the y-axes in the upper and lower panels. Error bars are standard errors of the mean. Vegetation treatments: PSC - plots consisting of peat, *Sphagnum* mosses and sedges (shrubs removed); PS - plots consisting of peat and *Sphagnum* (sedges and shrubs removed); P - plots consisting of bare peat (all vegetation removed). Intact plots consisted of peat, *Sphagnum* mosses, sedges and shrubs. Removal treatments were established in 2002.**

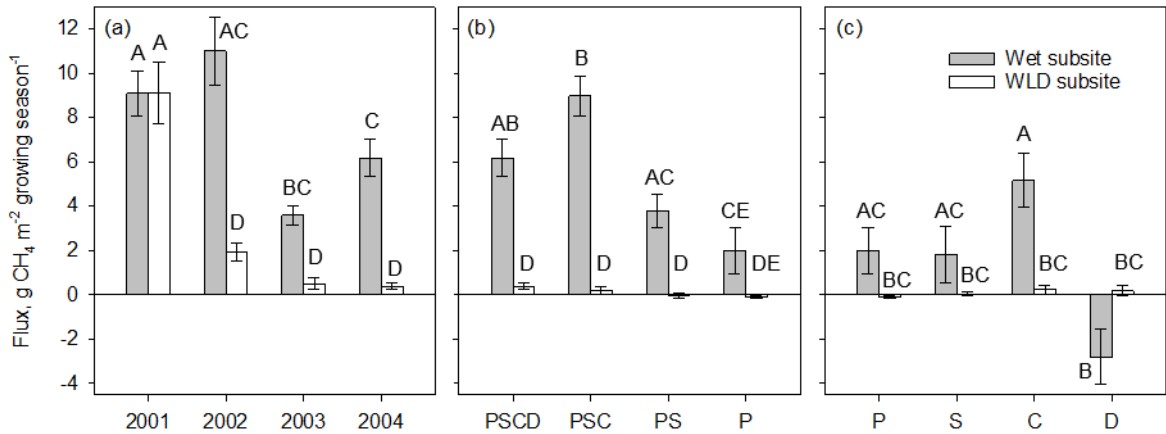

**Figure 3.** Seasonal (June-September) CH$_4$ flux (mean ± 1 standard error) in wet and water level (WLD) drawdown subsites a) in plots with intact vegetation (PSCD) during the four study years (2001 was a calibration year before the implementation of the WLD treatment), b) in different plant removal treatments plots in 2004 and c) by each plant group, the contribution of which to the total flux in 2004 was estimated from differences between the different plant removal treatments. Letters above bars denote differences among treatments, where bars with no letter in common are significantly different based on mixed-effects models presented in Table 2 (panels a and b) and based on two-way ANOVA test with Tukey pairwise comparisons (panel c). Plant removal treatments in (b): PSCD - plots with intact vegetation, consisting of peat, *Sphagnum* mosses, sedges and shrubs; PSC - plots consisting of peat, *Sphagnum* mosses and sedges (shrubs removed); PS - plots consisting of peat and *Sphagnum* mosses (shrubs and sedges removed), P - plots consisting of bare peat (all vegetation removed). Plant groups in (c): P – bare peat, S – *Sphagnum* mosses, C – sedges, D – dwarf shrubs.

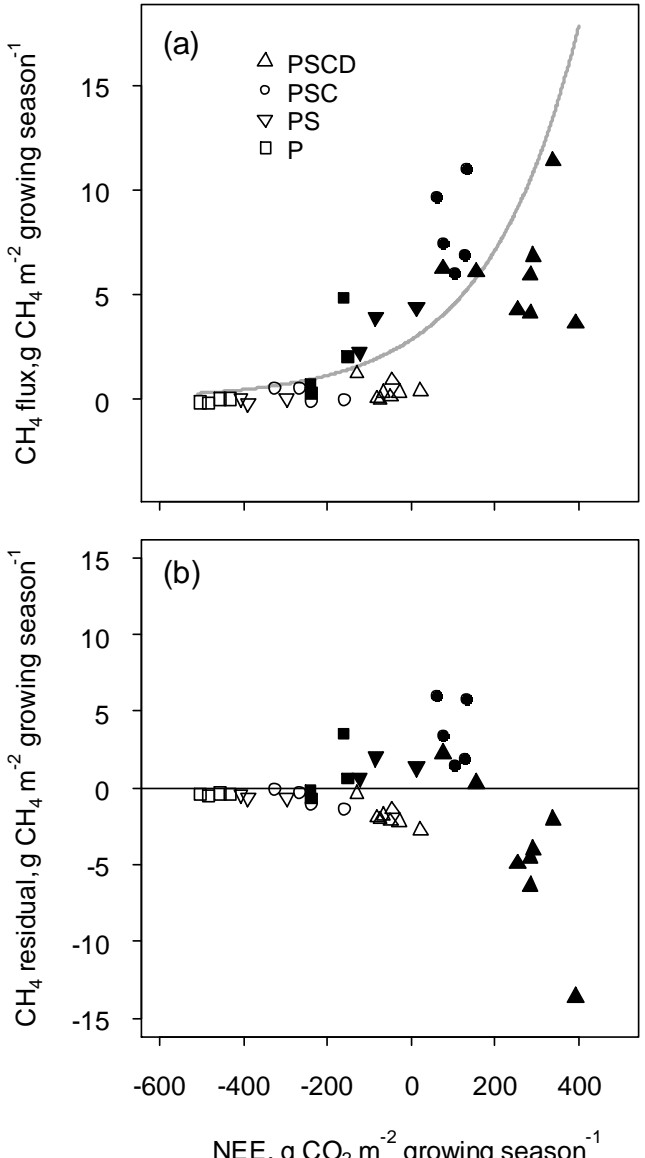

**Figure 4. a) The relationship between the net ecosystem CO₂ uptake (NEE) and CH₄ flux during the growing season 2004 described with an exponential model and b) the residuals of the model, in the different plant removal treatments in wet (solid symbols) and water level drawdown (open symbols) subsites. Vegetation treatments: PSCD - plots with intact vegetation, consisting of peat, *Sphagnum* mosses, sedges and shrubs; PSC - plots consisting of peat, *Sphagnum* mosses and sedges (shrubs removed); PS - plots consisting of peat and *Sphagnum* mosses (shrubs and sedges removed), P- plots consisting of bare peat (all vegetation removed). NEE is positive when the fen is a net sink of atmospheric CO₂. Methane flux is positive when the fen is a source of CH₄ to the atmosphere.**