# Peer review of "Interacting effects of vegetation components and water level on methane dynamics in a boreal fen"

_Biogeosciences, 2019_

## Referee Comment (RC1) · Tariq Munir (Referee) · 24 Oct 2019

General comments

Evaluation of the Interactive effects of plant functional groups and water table on CH4 fluxes in a boreal fen is exciting research and could confirm our understanding of the controls on CH4 fluxes in fen peatlands. Similar assessment studies have been conducted after the study years of 2001-2004. One of the strengths of research is that the emissions are partitioned based on vegetation components. This manuscript is concise and written very well with clarity and supports most of the earlier and later similar studies in the discussion section. The introduction covers relevant literature

and provides clear objectives that are achieved in results and aligned with conclusions. The paper merits publication once improved as per comments. The study results confirm many reported findings that water table level is the dominant control on CH4 fluxes, with vegetation components affect fluxes only under natural (or higher) water table level conditions. On the other hand, authors conclude that results are relevant for evaluating peatland CH4 flux responses to changing climatic conditions. I believe the authors could interpret the study conclusions carefully. To my analysis, these results are relevant for fen peatland (higher water table level) CH4 fluxes only. The results may not be applicable to bog peatland where water table level (in most cases) is deeper than Lakkasuo study fen (natural site) where mean CH4 fluxes decreased to zero (0.03 $\pm$ 0.03 CH4 m-2 month-1) after water table drawdown; Therefore authors may project the results relevancy to fen peatlands responding to changing climatic conditions. I notice that authors missed a significant opportunity of developing the CH4 emission factor for upscaling emissions for similar fen peatlands. The emission factors could be beneficial in reporting national or IPCC level CH4 emissions. Authors could look at Alm et al. 2007, Couwenberg and Fritz 2012, Levy et al. 2012 (GCB), Wilson et al. 2016, Strack et al. 2017 and few peatland CH4 studies from Western Canada. Study sites - Was the study site divided into two (wet or natural, and drier or WLD) in 2001 or 2002? It is given how far apart (radially) the two sites were, specifically, how far was the ditch from the wet site? Additionally, being the peatland complex (eccentric), did the authors verified if the two sites were similar in water table level and vegetation composition? These types of field investigations require additional (necessary) work so that the results obtained are solid. Was the ditch draining to some larger ditch/drain? Authors need to extend and clarify on sites, their chemistry and manipulation It would be methodologically challenging to create secluded vegetation removal treatments even after using paraffin wax, for example: • In PS, sedge stubbles/roots could still mediate fluxes • I believe that removal leaves underground roots/rhizomes, a large amount of substrate, which could result in undesirable data The authors need to explain how these problems were resolved. Based on earlier findings (for example, Conrad 2009, Hanson et al. 2000), they could support their removal treatments with several justifications – Lignin or associated polysaccharides are not but simpler carbohydrates or photosynthates are the dominant substrates. Clipping or removal disrupts the photosynthates movement to roots, which may not support dominant substrate-dependent CH4 production. . . . . . . . . . . . . . . . ..The explanations could also help discuss the water level $\times$ vegetation component interaction for CH4 fluxes The underlying mechanisms of CH4 production/release are established; however, authors need to briefly mention in the discussion to help the reader learn or refresh their understanding. The authors need to add some discussion (or sub-heading) on the water table level – vegetation interaction.

Specific comments

Line 13. The hyphen used here is inappropriate and could be replaced with a comma Line 14. Which growing seasons? Line 15. Insert "each of" after "of" I notice the use of super- or sub-scripts is inconsistent. Also, acronyms are not described in their first instances What could be the reasons the shrubs component attenuated the fluxes? References could be used for discussing ideas Line 22. What authors mean high here? Better say natural. Alternately, give how high? Line 23. Change "in" with "to" Line 24. Drawdown is a general term when mentioning climate change impacts; could be replaced with "deepening" Line 77. How the Lakkasuo peatland complex is an eccentric raised bog – a brief explanation would be helpful for the reader to understand how a nutrient-poor, oligotrophic fen existed within a bog. Line 81-87. Any visual/coverage estimates (numbers)? Line 100. I notice the use of spacing between a digit and a sign (- or +) is not consistent throughout the manuscript Line 102. Additional dot Line 110. Length $\times$ Width Line 124. Water table level Line 129. Any reference for species-specific Gaussian curves? Line 153-154. I notice authors tested here WL and Veg differences and provide results later in the results section) Line 238-241. Interesting to note that this study (2001-2004) compares results with earlier as well as later studies Figure 3. Add significance letters

Please also note the supplement to this comment:
https://www.biogeosciences-discuss.net/bg-2019-350/bg-2019-350-RC1-
supplement.pdf

———————————————————————

---

## Referee Comment (RC2) · Anonymous Referee #2 · 28 Oct 2019

This manuscript presents results for methane (CH4) flux from a water table drawdown and vegetation removal study conducted in an oligotrophic fen. All plots were studied for one year prior to any treatments and then the effects of water table lowering and vegetation removal were studied for the following three growing seasons. The authors observed that water table drawdown greatly reduced CH4 flux. In the first two years after treatment, vegetation removal plots often had higher fluxes than intact plots. By year three, plots with removal of dwarf shrubs continued to have higher fluxes than intact plots while plots with removal of shrubs and sedges or shrubs, sedges and Sphagnum had lower fluxes. Differences between vegetation treatments were only significant under wet conditions and not when the water table was lowered.

[Figure]

Overall, this study adds to our understanding of the interactions between water table and the presence of plant functional types on peatland CH4 emissions. However, I do think that the authors could add to the introduction and data analysis to better highlight how this paper moves beyond what we already know based on many of the studies they reference in this manuscript. In particular, I suggest that the authors add specific objectives, and possibly hypotheses, to better highlight the knowledge gap they aim to fill and how their study is unique in doing this. I also suggest that they consider the specific role of sedges more explicitly, potentially with a regression analysis between CH4 flux and sedge LAI, with interaction with water table. Some additional minor suggestions and further details on these revisions are given below.

Abstract: Just check the superscripts on the CH4 units and correct where necessary

Lines 27-28: Wetlands are the largest natural source of CH4, but much of this is from marshes, so I suggest adjusting this sentence. Also Saunois et al. 2016 is probably a better reference here than many that are given. Finally, I believe the correct reference for the first in the list is Mikaloff Fletcher et al. 2004, not Fletcher et al. 2004

Line 27: Here you use CH4, but later go back to using methane. I suggest you actually define CH4 here (so say methane (CH4)) and then use CH4 throughout the remainder of the manuscript.

Line 56: But what about trees? There is evidence they vent methane despite being shallow-rooted.

Line 69: "Fewer the roots" can just be "Fewer roots"

Lines 72-74: The introduction ends rather abruptly here and left me fairly unexcited about the study. I suggest that the authors could do a better job of highlighting the specific gap they are addressing here. Maybe also adding specific objectives and hypotheses would also help to transition to the methods here.

Line 194: There are two sentence here. Add a period or a connecting word.

Lines 199-200: Do you have data to show or a reference to another study to support this statement?

Lines 199-209: The differences between the years and the link to effects of plant removal and then stabilization seem to be largely conjecture. I agree that this makes sense, but without data to directly support how subsurface inputs were varying, and since weather and WT also varied between the years, I feel that some of the statements in these paragraphs are too definitive. I like how the changing patterns of fluxes are described, but unless there are direct observations to support "stabilization" in 2004, I'd suggest keeping the treatment effects for the discussion.

Line 229: Did you look at this pattern when the mean at each plot is considered? Since you have taken an average of all the plots for each treatment, this is not too different than looking at differences between plant removal treatments (e.g., Figure 3). Since you have so many replicates for each treatment type, it would be really nice to see how this relationship looks if each plot is a point on the graph in Figure 4. This could also help to illustrate the effect of PSCD being lower than the pattern driven by the other plots, which is currently a tough sell with only 4 points on the line for each water table treatment.

Lines 238-239: How did sedge cover differences in response to shrub removal affect the CH4 flux patterns? It would actually be interesting in general to see whether there was a correlation between sedge cover and CH4 flux when looking across all plots and whether there is an interaction with water table, particularly as this is alluded to in the introduction when reporting results of previous studies. I think this could be a really nice addition to the results and then could support this point made here.

Lines 268-284: Do you have any information from other studies at this study site to support this section. Even data on root distribution of the different species would help add confidence to this discussion.

Figure 2: I understand that the scale on the axes are kept the same on the top and

bottom row of plots so that they can be easily compared, but since the effect of water table drawdown on CH4 flux is already clearly shown in Figure 1 and the goal of this figure is to highlight vegetation effects, I suggest altering the scale on the bottom row so that variation between vegetation treatments can be seen. Since the fluxes are quite low post-water table drawdown, nothing can really be seen in this figure the way it is currently drawn. I would just point out the difference in axes in the caption and possibly even direct the reader back to Figure 1 for a clear comparison of control vs. water table drawdown fluxes.

Reference: Saunois M. et al. 2016. The global methane budget 2000 - 2012. Earth System Science Data, 8, 697-751.

---

## Author Comment (AC1) · 29 Nov 2019

General comments of the reviewer, author response below

Evaluation of the Interactive effects of plant functional groups and water table on CH4 fluxes in a boreal fen is exciting research and could confirm our understanding of the controls on CH4 fluxes in fen peatlands. Similar assessment studies have been conducted after the study years of 2001-2004. One of strengths of research is that the emissions are partitioned based on vegetation components. This manuscript is concise and written very well with clarity and supports most of the earlier and later similar studies in discussion section. Introduction covers relevant literature and provides clear objectives that are achieved in results and aligned with conclusions. The paper merits publication once improved as per comments.

The study results confirm many reported findings that water table level is the dominant control on CH4 fluxes, with vegetation components affect fluxes only under natural (or higher) water table level conditions. On the other hand, authors conclude that results are relevant for evaluating peatland CH4 flux responses to changing climatic conditions. I believe authors could interpret the study conclusions carefully. To my analysis, these results are relevant for fen peatland (higher water table level) CH4 fluxes only. The results may not be applicable to bog peatland where water table level (in most cases) is deeper than Lakkasuo study fen (natural site) where mean CH4 fluxes decreased to zero ($0.03 \pm 0.03$ CH4 m-2 month-1) after water table drawdown; Therefore authors may project the results relevancy to fen peatlands responding to changing climatic conditions. I notice that authors missed a significant opportunity of developing CH4 emission factor for upscaling emissions for similar fen peatlands. The emission factors could be beneficial in reporting national or IPCC level CH4 emissions. Authors could look at Alm et al. 2007, Couwenberg and Fritz 2012, Levy et al. 2012 (GCB), Wilson et al. 2016, Strack et al. 2017 and few peatland CH4 studies from Western Canada.

Study sites - Was the study site divided into two (wet or natural, and drier or WLD) in 2001 or 2002? It is given how far apart (radially) the two sites were, specifically, how far was the ditch from the wet site? Additionally, being the peatland complex (eccentric), did the authors verified if the two sites were similar in water table level and vegetation composition? These types of field investigations require additional (necessary) work so that the results obtained are solid.

Was the ditch draining to some larger ditch/drain? Authors need to extend and clarify on sites, their chemistry and manipulation

It would be methodologically challenging to create secluded vegetation removal treatments even after using paraffin wax, for example:

• In PS, sedge stubbles/roots could still mediate fluxes

• I believe that removal leaves underground roots/rhizomes, a large amount of substrate, which could result in undesirable data

The authors need to explain how these problems were resolved. Based on earlier findings (for example, Conrad 2009, Hanson et al. 2000), they could support their removal treatments with several justifications – Lignin or associated polysaccharides are not but simpler carbohydrates or photosynthates are the dominant substrates. Clipping or removal disrupts the photosynthates movement to roots, which may not support dominant substrate-dependent CH4 production. The explanations could also help discuss the water level $\times$ vegetation component interaction for CH4 fluxes

The underlying mechanisms of CH4 production/release are established; however, authors need to briefly mention in the discussion to help the reader learn or refresh their understanding. The authors need to add some discussion (or sub-heading) on the water table level – vegetation interaction.

> We thank the reviewer for the positive overall statement about our manuscript and the comments that helped us to improve our manuscript.

> First, we have now specified in the Abstract and Conclusions that our results are applicable for fen peatlands. Second, we have added the suggested $CH_4$ emission factor calculation, please see L178-179 and L229-230.

> The description of the study site and experimental design have now been modified so that all requested information should be more easily found in the text, including pH and more specific information about the water level manipulation (L100-101, L114-120).We would like to thank the reviewer for the interesting insights related to the vegetation manipulation used in this study. We have now added for example a new regression model describing the relationship between sedge leaf area and $CH_4$ flux (from L195 onwards) as well as extended the description of the methods (from L99 onwards). We would like to keep the original subheading "Water level regulates the role of the vegetation" in the discussion instead of mentioning interaction in the subheading. However, this section has now been extended to include discussion about the new regression model (L364-371).

> We agree that after vegetation removal treatments the sedge stubble and roots could still act as conduits for $CH_4$, and the dead roots and rhizomes could provide substrate for methanogenesis. We have discussed these aspects in Discussion in the section "Delay in the plant removal treatment effect". Based on the literature, and our own observation of stable fluxes and no more sedge regrowth, it seems that the substrate supply from the decaying roots and rhizomes was exhausted, and the aerenchymatous pathway disappeared, by the 3rd year of the vegetation manipulations. Our quantitative results on the contribution of the vegetation components to the fluxes is based on that year's (2004) data.

> We have also discussed the points mentioned by the reviewer of the tissues of high lignin content being less favoured substrate for methanogenesis and having a positive relationship with methane oxidation rates (L319-322), and that the photosynthates and fresh carbon compounds transported through the roots are the main substrate for methane production (L42-50).

Specific comments, author response indented below each specific comment

Line 13. The hyphen used here is inappropriate and could be replaced with a comma

> Replaced as suggested.

Line 14. Which growing seasons?

> Specified as suggested (L13).

Line 15. Insert "each of" after "of"

We thank for the comment but think the sentence is more concise without the addition.

I notice the use of super- or sub-scripts is inconsistent. Also, acronyms are not described in their first instances

Super- and sub-scripts as well as acronyms have now been checked throughout the text.

What could be the reasons the shrubs component attenuated the fluxes? References could be used for discussing ideas

We have added discussion on this matter in addition to the existing discussion from L285 onwards.

Line 22. What authors mean high here? Better say natural. Alternately, give how high?

Changed as suggested (L22).

Line 23. Change "in" with "to"

Changed as suggested (L23).

Line 24. Drawdown is a general term when mentioning climate change impacts; could be replaced with "deepening"

We understand the point of the reviewer, but would like to keep the word 'drawdown' here as it has been used widely in this context (e.g Strack et al. 2007, Freeman et al. 2012, Kokkonen et al. 2019).

Line 77. How the Lakkasuo peatland complex is an eccentric raised bog – a brief explanation would be helpful for the reader to understand how a nutrient-poor, oligotrophic fen existed within a bog.

We have specified the description on L95 and L100.

Line 81-87. Any visual/coverage estimates (numbers)?

Cover estimates have now been added as suggested (from L 102 onwards).

Line 100. I notice the use of spacing between a digit and a sign (- or +) is not consistent throughout the manuscript

These have now been checked.

Line 102. Additional dot

Corrected

Line 110. Length $\times$ Width

Specified as suggested (L139).

Line 124. Water table level

Based on literature this is an issue of personal taste and in here, and in most of the works by our group, we have decided to use term water level. We have now made sure throughout the manuscript that the use of the term is consistent.

Line 129. Any reference for species-specific Gaussian curves?

    A reference has been now cited in the sentence (L157).

Line 153-154. I notice authors tested here WL and Veg differences and provide results later in the results section)

Line 238-241. Interesting to note that this study (2001-2004) compares results with earlier as well as later studies

    We have tried to include the most relevant references.

Figure 3. Add significance letters

    We have now added the letters.

---

## Author Comment (AC2) · 29 Nov 2019

General comments of the reviewer, author response below

This manuscript presents results for methane (CH4) flux from a water table drawdown and vegetation removal study conducted in an oligotrophic fen. All plots were studied for one year prior to any treatments and then the effects of water table lowering and vegetation removal were studied for the following three growing seasons. The authors observed that water table drawdown greatly reduced CH4 flux. In the first two years after treatment, vegetation removal plots often had higher fluxes than intact plots. By year three, plots with removal of dwarf shrubs continued to have higher fluxes than intact plots while plots with removal of shrubs and sedges or shrubs, sedges and Sphagnum had lower fluxes. Differences between vegetation treatments were only significant under wet conditions and not when the water table was lowered.

Overall, this study adds to our understanding of the interactions between water table and the presence of plant functional types on peatland CH4 emissions. However, I do think that the authors could add to the introduction and data analysis to better highlight how this paper moves beyond what we already know based on many of the studies they reference in this manuscript. In particular, I suggest that the authors add specific objectives, and possibly hypotheses, to better highlight the knowledge gap they aim to fill and how their study is unique in doing this. I also suggest that they consider the specific role of sedges more explicitly, potentially with a regression analysis between CH4 flux and sedge LAI, with interaction with water table. Some additional minor suggestions and further details on these revisions are given below.

> We thank the reviewer for the valuable comments that helped us to significantly improve our manuscript. As suggested we objectives and hypothesis on L80 onwards to clarify the aim of our work, and those have now been added. Further, we added the requested regression analysis between $CH_4$ flux and sedge LAI and the interaction with water table. Please see the extended analysis explained from L195 onwards with results from L276 onwards.

Specific comments, author response indented below each specific comment

Abstract: Just check the superscripts on the CH4 units and correct where necessary

> These have now been checked and corrected where necessary.

Lines 27-28: Wetlands are the largest natural source of CH4, but much of this is from marshes, so I suggest adjusting this sentence. Also Saunois et al. 2016 is probably a better reference here than many that are given. Finally, I believe the correct reference for the first in the list is Mikaloff Fletcher et al. 2004, not Fletcher et al. 2004

> We clarified the sentence and added Saunois et al. 2016 as reference. We also removed some of the references, including Mikaloff Fletcher.

Line 27: Here you use CH4, but later go back to using methane. I suggest you actually define CH4 here (so say methane (CH4)) and then use CH4 throughout the remainder of the manuscript.

> We have now made the suggested changes.

Line 56: But what about trees? There is evidence they vent methane despite being shallow-rooted.

> It is true that trees have been found to transport notable quantities of methane in tropical peatlands, although to our knowledge such studies have not yet been

published from boreal peatlands. Our site is an open fen, but it is worth mentioning in the introduction the potential role of trees in transporting methane. We have now modified L59-60.

Line 69: "Fewer the roots" can just be "Fewer roots"

We have made the suggested change (L76).

Lines 72-74: The introduction ends rather abruptly here and left me fairly unexcited about the study. I suggest that the authors could do a better job of highlighting the specific gap they are addressing here. Maybe also adding specific objectives and hypotheses would also help to transition to the methods here.

We agree with the reviewer and have now added objectives and hypotheses addressing the specific gap.

Line 194: There are two sentence here. Add a period or a connecting word.

We made the suggested modifications.

Lines 199-200: Do you have data to show or a reference to another study to support this statement?

Please see the answer to the next comment.

Lines 199-209: The differences between the years and the link to effects of plant removal and then stabilization seem to be largely conjecture. I agree that this makes sense, but without data to directly support how subsurface inputs were varying, and since weather and WT also varied between the years, I feel that some of the statements in these paragraphs are too definitive. I like how the changing patterns of fluxes are described, but unless there are direct observations to support "stabilization" in 2004, I'd suggest keeping the treatment effects for the discussion.

We have now deleted the first sentence of both of these two paragraphs (L240 and L249) and kept only the information that can be clearly justified by the results of this study. The treatment artefacts and stabilization thereafter have now only been dealt with in the discussion.

Line 229: Did you look at this pattern when the mean at each plot is considered? Since you have taken an average of all the plots for each treatment, this is not too different than looking at differences between plant removal treatments (e.g., Figure 3). Since you have so many replicates for each treatment type, it would be really nice to see how this relationship looks if each plot is a point on the graph in Figure 4. This could also help to illustrate the effect of PSCD being lower than the pattern driven by the other plots, which is currently a tough sell with only 4 points on the line for each water table treatment.

This figure has now been redrawn according to the suggestions.

Lines 238-239: How did sedge cover differences in response to shrub removal affect the CH4 flux patterns? It would actually be interesting in general to see whether there was a correlation between sedge cover and CH4 flux when looking across all plots and whether there is an interaction with water table, particularly as this is alluded to in the introduction when reporting results of previous studies. I think this could be a really nice addition to the results and then could support this point made here.

We thank the reviewer for the suggestion and have now conducted the suggested analysis. The methods have been explained from L195 onwards with results from L276 onwards.

Lines 268-284: Do you have any information from other studies at this study site to support this section. Even data on root distribution of the different species would help add confidence to this discussion.

We added here (L344-345) two references to root biomass studies at the same peatland complex than in this study and a nearby bog (Mäkiranta et al. 2018 and Korrensalo et al. 2018).

Figure 2: I understand that the scale on the axes are kept the same on the top and bottom row of plots so that they can be easily compared, but since the effect of water table drawdown on CH4 flux is already clearly shown in Figure 1 and the goal of this figure is to highlight vegetation effects, I suggest altering the scale on the bottom row so that variation between vegetation treatments can be seen. Since the fluxes are quite low post-water table drawdown, nothing can really be seen in this figure the way it is currently drawn. I would just point out the difference in axes in the caption and possibly even direct the reader back to Figure 1 for a clear comparison of control vs. water table drawdown fluxes.

The figure has now been redrawn according to the suggestion.

Reference: Saunois M. et al. 2016. The global methane budget 2000 - 2012. Earth System Science Data, 8, 697-751.

---

## Author Response (AR2)

We thank the Reviewer and the Editor for the time used with our manuscript "Interacting effects of vegetation components and water level on methane dynamics in a boreal fen" (bg-2019-350) and the positive statement regarding our work. We have now completed the final remarks of the reviewer as listed below.

On behalf of all authors,

Aino Korrensalo

10   Suggestions for revision or reasons for rejection (will be published if the paper is accepted for final publication)

Line 89: Should be "a smaller proportion"
    Added as suggested, now on L99.

Lines 101-104: The edits here are fine, but now this sentence doesn't quite make sense. Please reword.
    Reworded to improve clarity. Please see L113.

20   Lines 323 and 365: subscript needs on CH4
    Changed as suggested, see L323.

[revised manuscript text omitted]